# Household indebtedness and well-being: Evidence from Australia

**Maram Tammam, Khaled Toffaha** ◉**\*, Michael A. Kortt** ◉**, Albert Wijeweera**

Department of Management Science & Engineering, Khalifa University of Science and Technology, Abu Dhabi, United Arab Emirates

\* Khaled.mtoffaha@ku.ac.ae

## Abstract

This study examines the relationship between household debt and health among Australian adults aged 18-65, utilising five waves of the Household, Income and Labour Dynamics in Australia (HILDA) Survey (2006, 2010, 2014, 2018, and 2022). The analysis distinguishes between unsecured credit card debt and secured mortgage debt, examining outcomes including health satisfaction, mental health (as measured by the SF-36 Mental Component Score (MCS)), obesity, and repayment stress. In individual fixed-effects models, each 10-percentage-point increase in the credit-card debt-to-income ratio is associated with a 0.028-point decline in health satisfaction and a 0.402-point decline in the MCS; mortgage-debt estimates are small and inconsistent. Reporting mortgage-repayment stress corresponds to a 0.081-point lower health satisfaction and a 2.884-point lower MCS. Evidence for obesity is weak in the fixed-effects models. Overall, the findings suggest that the type of debt and the stress it generates are more significant for health than the total amount borrowed.

## 1 Introduction

Household debt has become a significant concern in advanced economies. It matters not only for its macroeconomic implications but also for its potential effects on individual well-being [1–3]. Within this broader financial context, the growth of unsecured credit-card debt [4,5] and the substantial financial commitments associated with mortgage borrowing [6,7] warrant closer scrutiny, as these two forms of debt differ in their structure and risk profiles. In Australia and internationally, the recent combination of rising interest rates and persistent inflation has heightened concerns about how such liabilities influence health outcomes both directly through financial stress and indirectly through long-term strain on household resources [8–10].

Australia ranks among the highest OECD countries in terms of household debt-to-income ratios, with mortgage borrowing accounting for the largest share of household liabilities [3]. Debt burdens are particularly acute for vulnerable groups: 29% of low-income households are classified as over-indebted (debt exceeding three times income), placing Australia sixth among 23 OECD economies [11]. High-cost credit

**Data availability statement:** Data cannot be shared publicly because access to the Household, Income and Labour Dynamics in Australia (HILDA) Survey is restricted under the licensing conditions set by the Melbourne Institute of Applied Economic and Social Research. Data are available from the Melbourne Institute (contact via https://melbourneinstitute.unimelb.edu.au/hilda) for researchers who meet the criteria for access to confidential data.

**Funding:** This work was supported by Khalifa University. The funder had no role in study design, data collection and analysis, decision to publish, or preparation of the manuscript.

**Competing interests:** No authors have competing interests.

also compounds financial strain: as of 2023, average standard credit card interest rates remained close to 20%, and Australian Securities and Investments Commission (ASIC) reported in 2018 that nearly one in five cardholders met at least one indicator of problematic debt, such as persistent balances, overdue accounts, or hardship support [12–14].

Mortgage stress has also escalated. In April 2024, almost one-third of mortgage holders were considered at risk [15]. Overall, household debt has risen from approximately 80% of GDP in the early 2000s to nearly 112% by late 2024, surpassing the OECD average [16]. While prior research has highlighted these broad economic trends, relatively few studies have explicitly compared the health consequences of different types of household debt.

Building on Keese and Schmitz [17], who used German panel data to demonstrate that repayment difficulties with consumer credit and mortgage loans undermine health, the present study takes a different approach. Rather than focusing solely on repayment problems, we analyse stock measures of household liabilities – outstanding balances recorded at the interview date – which capture levels of indebtedness rather than flows (e.g., new borrowing or missed repayments). Specifically, we examine credit-card balances and mortgage debt to assess how overall indebtedness relates to well-being. This distinction is important: repayment problems capture acute financial strain, while stock debt reflects longer-term exposure to financial obligations. Applying this framework to the Australian context offers a fresh perspective on how debt structure impacts health.

A substantial body of research has documented the adverse consequences of financial strain on health, linking debt burdens to poorer self-reported health and heightened psychological distress [18]. Additionally, financial strain has been associated with physical ailments, including cardiovascular problems and weakened immune function [19–21]. However, evidence remains limited on how specific types of debt, such as credit card balances compared to long-term mortgage obligations, differentially affect health outcomes. Credit card debt, with its high interest rates and revolving balances, may engender more acute psychological strain than mortgage debt, which is long-term, structured, and asset-backed [6,22,23].

This study addresses this gap by analysing the impact of credit card and mortgage debt on three dimensions of well-being: overall health satisfaction, mental health, and physical health, while also considering the stress associated with mortgage repayment difficulties. It advances the literature in several ways. First, it draws on the nationally representative Household, Income and Labour Dynamics in Australia (HILDA) Survey. This rich panel dataset follows individuals over time, providing detailed measures of financial circumstances, health outcomes, and socioeconomic factors. This setting enables a more robust analysis of the dynamic relationships between debt and health status. Second, household debt is disaggregated into credit card and mortgage components, allowing for an assessment of how different liabilities impact well-being. Third, to address unobserved heterogeneity, fixed-effects (FE) models are employed, controlling for time-invariant individual characteristics that

may confound the estimates. Fourth, lagged debt variables are incorporated to capture delayed health effects, recognising that stress from financial burdens may accumulate over time. Finally, three dimensions of health are examined: (i) self-reported health satisfaction, (ii) mental health as measured by the SF-36 index, and (iii) physical health, proxied by obesity, along with a direct indicator of mortgage stress, the inability to meet mortgage payments. Together, this multidimensional approach offers a comprehensive evaluation of how debt impacts individual well-being in Australia.

The remainder of this paper is organised as follows. Sect 2 reviews the existing literature on the relationship between debt and health outcomes. Sect 3 describes the data sources and empirical strategy. Sect 4 presents the results, and Sect 5 offers a discussion of the findings. Sect 6 concludes the paper.

## 2 Related work

Socio-economic status (SES) is widely recognised as a fundamental determinant of health, with lower SES associated with increased morbidity, psychological distress, and reduced life expectancy [24–26]. Financial strain, particularly in the form of household debt, has emerged as a key mediator in this relationship, linking financial insecurity with mental and physical health risks. Following the Global Financial Crisis, debt has become a persistent feature of many advanced economies, driven by the need to sustain consumption and manage financial shocks [2,27]. As a result, household indebtedness is increasingly viewed not only as an economic concern but also as a significant public health issue [28,29].

A substantial body of research shows that debt burdens are associated with poorer self-reported health, heightened psychological distress, and even physical ailments such as cardiovascular disease and weakened immune function [19–22]. Stress arising from financial insecurity is understood as a key mechanism linking debt to both mental and physical health outcomes [30]. Sweet et al. (2013), for example, demonstrated that high financial debt relative to assets is associated with magnified perceived stress, increased depression, and poorer general health, even after controlling for prior SES and health [21].

However, much of this research treats debt as an aggregate construct, masking potentially different effects of specific liabilities. Recent studies increasingly distinguish between unsecured debt, such as credit card balances, and secured debt, such as mortgages. Unsecured debt is often found to be especially stressful due to high interest rates, revolving balances, and repayment pressure [31]. The revolving nature of credit card borrowing, often used to finance daily expenses or emergencies, can trap households in cycles of indebtedness and exacerbate psychological strain. Hojman et al. (2016) confirm this by showing that persistent over-indebtedness, particularly non-mortgage debt, is strongly associated with depressive symptoms [32].

By contrast, the evidence on mortgage debt in health research is more mixed. On the one hand, it is secured against an asset and traditionally associated with wealth accumulation and stability; on the other, high mortgage burdens create acute financial stress, especially during housing market downturns. Struggles with repayments and foreclosure have been linked to elevated psychological distress and adverse physical health outcomes [33,34]. Clayton et al. (2015) further highlight that while short- or medium-term debt may buffer households against financial shocks, long-term mortgage and unsecured debt are associated with poorer health [35]. This highlights that the type and duration of liabilities are crucial for health outcomes.

More recent work extends this literature to vulnerable groups such as older adults, who often face declining incomes alongside rising health risks. Using data from the English Longitudinal Study of Ageing, Hiilamo (2024) found that debt is significantly associated with worse outcomes across both mental and physical health [36]. Mudrazija and Butrica (2023), drawing on U.S. panel data, reported that unsecured debt has a powerful adverse effect on older adults' well-being [37]. Sweet (2020) also shows that debt-induced trade-offs, such as postponing medical care or limiting essential spending, directly harm self-rated health while increasing anxiety and depression [38].

Despite this growing evidence, much of the literature remains concentrated in North America and Europe. Fewer studies have examined other high-income settings such as Australia, where debt-to-income ratios rank among the highest

in the OECD and mortgage borrowing constitutes the largest share of household liabilities [39]. Bentley et al.(2022) provide Australian evidence linking housing unaffordability to elevated psychological distress, particularly among younger households, but comprehensive longitudinal evidence on debt and health remains scarce [40].

In sum, existing research demonstrates that debt is detrimental to health, but often fails to distinguish the heterogeneous effects of specific liabilities or adequately address bidirectional causality between debt and health. In line with Keese and Schmitz (2014), who highlighted the health effects of repayment difficulties across consumer credit and mortgage liabilities, this study extends their approach by focusing on stock measures of household debt in the Australian context [17]. By disaggregating credit card balances and mortgage liabilities, examining how the level of indebtedness itself, rather than repayment strain alone, relates to health satisfaction, mental health, and physical health outcomes.

## 3 Data and empirical strategy

### 3.1 Data source

This study draws on the HILDA Survey, a nationally representative panel survey initiated in 2001, which follows Australian individuals and households, collecting rich information on their socioeconomic, demographic, and health characteristics. The analysis draws on waves 6, 10, 14, 18, and 22 (2006-2022). These waves were selected to maintain consistency in variable definitions, include the SF-36 Mental Component Score, and allow sufficient temporal spacing to observe meaningful variation in debt and health outcomes. This 16-year window also spans major macroeconomic shifts, including the aftermath of the Global Financial Crisis, varying interest rate environments, and evolving debt patterns. The final analytic sample comprises adults aged 18-65, yielding an unbalanced panel with 10,874 observations in 2006, 11,316 in 2010, 14,525 in 2014, 14,363 in 2018, and 12,991 in 2022. The age restriction ensures focus on the working-age population most likely to hold credit card and mortgage debt.

### 3.2 Health outcomes

To capture the multidimensional nature of health and well-being, three established measures available in the HILDA Survey are drawn upon. Health satisfaction is based on a self-reported five-point scale (1 = completely dissatisfied to 5 = completely satisfied) in response to the question "*How satisfied are you with your health*?" This subjective assessment has been shown to predict both current and future health outcomes [41]. Mental health is measured using the SF-36 Mental Component Score (MCS), a validated scale ranging from 0 to 100 that covers vitality, social functioning, emotional role limitations, and overall psychological well-being, and is widely used in population health research [42]. Obesity is defined as a binary indicator equal to 1 for individuals' BMI $\geq$ 30 kg/m² (calculated from self-reported height and weight) and zero otherwise. Despite the possibility of reporting error, validation studies confirm that HILDA's self-reported anthropometric data are reasonably accurate [43]. To aid comparability across continuous outcomes, we also report specifications in which health satisfaction and MCS are standardised to mean 0 and standard deviation 1, so coefficients can be interpreted in SD units. Together, these measures provide complementary subjective, psychometric, and objective indicators of health and well-being.

### 3.3 Debt measures

Household indebtedness is captured using stock-based indicators and a short-term repayment measure. The credit card debt-to-income (cDTI) ratio is defined as outstanding credit card balances divided by annual household income (CPI-adjusted to 2022 dollars). Credit card borrowing represents unsecured, high-interest revolving credit, often used for consumption smoothing or emergencies, and the cDTI formulation allows comparisons across income levels. The mortgage debt-to-income (mDTI) ratio is calculated as the outstanding mortgage divided by household income. The DTI ratios are expressed as proportions, where 0.1 = 10%, 0.5 = 50%, and 1.0 = 100% of annual household income. Mortgage debt is secured against housing assets and typically constitutes the largest share of household liabilities, though high balances

can generate repayment strain. To limit the impact of outliers, cDTIs are trimmed at 0.5 and mDTIs at 2.0, thresholds informed by the data distribution and economic plausibility. Robustness checks confirm that the results are not sensitive to alternative cut-offs. Additionally, model robustness was further assessed by trimming mortgages at 6 and 10, with conclusions remaining stable.

To facilitate interpretation, all debt-to-income ratios were divided by 10 prior to estimation; coefficients can be read as the change in the outcome associated with a 10 percentage point increase in the relevant ratio. This scaling is applied consistently across pooled OLS and fixed effects specifications.

Alongside these stock measures, financial stress is included, defined as a binary indicator of whether a household reported being unable to meet mortgage repayments on time in the past year. This measure captures acute repayment distress and provides a behavioural complement to the debt ratios. Because the financial stress measure was not collected in wave 10, analyses incorporating this variable rely on a reduced sample. Together, the DTI ratios and stress indicator provide a multifaceted view of household debt, encompassing both overall indebtedness and the likelihood of immediate repayment difficulty.

### 3.4 Control variables

To account for the broad range of socioeconomic and demographic factors that may influence household debt and health outcomes, the analysis incorporates a comprehensive set of controls. Age (measured in years) and years of education capture life-cycle and human capital differences, with a separate dummy for missing educational responses. Demographic characteristics include sex (1 = female; 0 = otherwise) and three separate marital status dummies: married (1 = married; 0 = otherwise), divorced (1 = divorced; 0 otherwise), and widowed (1 = widowed; 0 otherwise). Indigenous status (1 = Indigenous; 0 = otherwise) and country of birth (1 = born overseas; 0 = otherwise) are also included.

Family structure is captured by the presence of children (1 = has children; 0 = otherwise), along with a dummy for missing child information. Labour-market status is measured using a categorical variable that distinguishes between employed, unemployed, and not in the labour force, with employed serving as the reference category. Real household income (CPI-adjusted to 2022 dollars) is also included to control for differences in financial capacity.

Health-care access is proxied by private health insurance status (1 = has private insurance; 0 = otherwise), together with a dummy for missing responses. Geographic variation is captured by an indicator for residence in a major city (1 = city; 0 = otherwise). Year dummies for 2010, 2014, 2018, and 2022 (with 2006 as the reference) are also included to absorb macroeconomic shocks and time-specific influences, consistent with the standard practice of including time fixed effects.

### 3.5 Empirical strategy

The empirical analysis exploits the panel structure of the HILDA Survey to estimate the relationship between household debt and health outcomes across five survey waves (2006, 2010, 2014, 2018, and 2022). The models take the general form:

$$Y_{it} = \alpha + \beta_1 CreditCard\_DTI_{it} + \beta_2 Mortgage\_DTI_{it} + \mathbf{X}_{it}\gamma + \delta_t + \epsilon_{it}$$

where $Y_{it}$ represents health outcomes for individual $i$ at time $t$, $CreditCard\_DTI_{it}$ and $Mortgage\_DTI_{it}$ are the respective debt-to-income ratios, $\mathbf{X}_{it}$ is a vector of control variables, $\delta_t$ captures survey wave fixed effects, and $\epsilon_{it}$ is the error term. In pooled OLS specifications, $\alpha$ is the common intercept. In the fixed-effects (FE) models $\alpha$ is replaced with $\alpha_i$, capturing time-invariant individual heterogeneity that could otherwise confound the estimates.

Pooled OLS models are first employed as a baseline, providing an initial benchmark for the associations between debt and health outcomes across individuals and survey waves. However, OLS estimates are vulnerable to bias from unobserved heterogeneity, such as personality traits that might affect debt accumulation and health. To address this concern,

individual fixed-effects models are estimated to control for all unobserved time-invariant characteristics. This within-person design allows us to identify how changes in debt within the same individuals over time are associated with changes in their health outcomes. For the pooled obesity model, results are presented as average marginal effects in percentage points per ten percentage point increase in the debt to income ratio, with 95% confidence intervals accompanying all estimates.

For the obesity outcome, logistic regression is employed in the pooled OLS specification. In the FE models, linear probability models are used instead, as conditional logit estimation encountered convergence difficulties in the panel setting. The LPM approach ensures that the FE estimates remain stable and interpretable, and results are robust to alternative specifications.

To further probe the robustness of the FE results, two key variants are estimated. First, the sample is restricted to individuals who were continuously employed across their observed waves (i.e., 'always-working FE'). This addresses concerns that employment shocks might drive indebtedness and health deterioration, potentially biasing estimates. By focusing on those with stable labour force attachment, the persistence of the debt-health relationship independent of employment instability can be assessed. Second, lagged FE models are estimated in which current health outcomes are regressed on debt measures from the previous wave. This specification recognises that financial stress may take time to manifest in health outcomes and helps mitigate concerns about reverse causality, such as deteriorating health leading households to borrow more. Evidence of consistent lagged effects would strengthen the interpretation of debt as a precursor to adverse health outcomes.

Finally, while the primary focus is on debt-to-income ratios, sensitivity checks were also performed. Results are qualitatively similar when debt variables are expressed in logarithmic form, and findings are not sensitive to alternative trimming thresholds for extreme debt-to-income ratios. These diagnostics aid interpretation but are not strictly comparable across model families. This reinforces the robustness of the empirical design and ensures that results are not artifacts of variable construction. Taken together, the combination of pooled OLS, individual FE, always-working FE, and lagged FE specifications provides a layered and credible approach to assessing the debt-health relationship.

## 4 Results

### 4.1 Descriptive statistics

Before turning to the regression analysis, it is useful to examine broad descriptive patterns in household debt and health outcomes. Tables 1–3 provide complementary perspectives: temporal trends in debt (Table 1), differences in health outcomes by debt type (Table 2), and subgroup variation by employment stability (Table 3). Together, they illustrate the diverse ways in which household indebtedness interacts with economic capacity and well-being.

Table 1 demonstrates striking declines in unsecured debt over time. Mean credit card debt among all households fell from $2,810 in 2006 to just $767 in 2022 (real 2022 AUD), while the share of households carrying such debt dropped from 35.1% to 15.2%. Among households with credit card debt, average balances declined from $8,000 to $5,055 across the same period, and debt-to-income ratios fell from 7.4% to 4.2%. These figures suggest both reduced reliance on credit card financing and more cautious household borrowing behaviour. By contrast, mortgage debt exhibited stability at a high level. Mean mortgage debt across all households increased from $117,587 in 2006 to $179,588 in 2022, a rise of more than 50% in real terms. Mortgage participation rates held steady at around 44-46% of households, and among mortgage holders, debt-to-income ratios consistently exceeded 100%, peaking at 106% in 2018 before easing slightly to 103% in 2022. The proportion of households reporting late mortgage payments fluctuated only modestly between 6.5% and 7.0% where data were collected. Overall, the data highlight a divergence: credit card debt has decreased sharply in prevalence and size, whereas mortgage debt has remained substantial and entrenched.

Table 2 shifts the focus to health outcomes, and here the contrasts are particularly striking. The average health satisfaction in the full sample is 3.42 (on a 1-5 scale), but it falls to 3.37 among those with credit card debt, while rising slightly

**Table 1**. Summary statistics for household debt and financial stress (Real 2022 AUD).

| Measure | 2006 | 2010 | 2014 | 2018 | 2022 |
|---|---|---|---|---|---|
| Mean credit card debt (all HHs) in AUD | $2,810 | $3,264 | $2,320 | $1,781 | $767 |
| Credit card debt-to-income (all HHs) | 2.54% | 2.44% | 1.92% | 1.43% | 0.63% |
| Mean credit card debt (debtors only) in AUD | $8,000 | $9,585 | $7,966 | $7,244 | $5,055 |
| Credit card debt-to-income (debtors only) | 7.4% | 7.3% | 6.6% | 5.9% | 4.2% |
| % with credit card debt | 35.1% | 34.1% | 29.1% | 24.6% | 15.2% |
| Mean mortgage debt (all HHs) in AUD | $117,587 | $145,810 | $152,192 | $171,361 | $179,588 |
| Mortgage debt-to-income (all HHs) | 28.5% | 29.0% | 27.8% | 26.2% | 27.2% |
| Mean mortgage debt (debtors only) in AUD | $267,107 | $316,636 | $338,944 | $380,293 | $394,292 |
| Mortgage debt-to-income (debtors only) | 101.8% | 103.0% | 105.1% | 106.0% | 102.9% |
| % with mortgage debt | 44.0% | 46.0% | 44.9% | 45.1% | 45.5% |
| % could not pay mortgage on time | 7.0% | — | 6.8% | 6.5% | 6.5% |
| Observations | 10,874 | 11,316 | 14,525 | 14,363 | 12,991 |

Notes: "Debtors only" includes households with positive debt values. "Could not pay mortgage on time" was not collected in 2010. The sample consists of respondents aged 18 to 65. Credit card debt-to-income ratios are trimmed at 0.5, and mortgage debt-to-income ratios at 2 to reduce the influence of extreme outliers.

**Table 2**. Sample means of health measures by household debt status.

| Health Measure | Full Sample | HH with Credit Debt | HH with Mortgage Debt | Observations |
|---|---|---|---|---|
| Health satisfaction (1–5) | 3.42 | 3.37 | 3.50 | 53,605 |
| MCS (0–100) | 72.26 | 72.13 | 73.94 | 54,138 |
| Obesity (%) | 25.76% | 30.27% | 24.75% | 52,058 |

Notes: Health satisfaction is based on self-reported ratings on a 1–5 scale. Mental Component Score (MCS) is derived from the SF-36 health survey. Obesity is defined as a BMI ≥ 30. Values reflect the proportion of individuals classified as obese. The sample includes respondents aged 18 to 65 years.

**Table 3**. Descriptive statistics for subgroups by employment status.

| Measure | Always Employed | Not Always Employed |
|---|---|---|
| Mean credit card debt (all households) | $2,358 | $1,776 |
| Mean credit card debt (debtors only) | $7,907 | $7,771 |
| % with credit card debt | 29.8% | 22.8% |
| Mean mortgage debt (all households) | $187,908 | $105,909 |
| Mean mortgage debt (debtors only) | $359,106 | $310,022 |
| % with mortgage debt | 52.3% | 34.2% |
| % could not pay the mortgage on time | 5.2% | 8.9% |
| Health satisfaction (1–5 scale) | 3.57 | 3.19 |
| Mental Component Score (0–100) | 74.44 | 69.01 |
| Obesity (%) | 23.5% | 29.2% |
| Observations | 37,157 | 24,422 |

Notes: Figures in AUD are adjusted to 2022 real values. "Always Employed" refers to individuals continuously employed across observed waves (i.e., never reported being unemployed or not in the labor force). Mental Component Score (MCS) is derived from the SF-36 health survey. Higher values indicate better mental health.

to 3.50 among those with mortgage debt. Similarly, mean Mental Component Scores (MCS) are lowest for credit card debtors (72.13), highest among mortgage holders (73.94), and higher for the full sample (72.26). Obesity rates exhibit a reverse pattern: 30.3% among credit card debtors, compared to 25.8% in the full sample and 24.8% among mortgage debtors. These figures indicate that unsecured debt is associated with systematically worse health outcomes, whereas

mortgage debtors appear to enjoy modestly better outcomes than the average. This divergence may reflect the different socioeconomic profiles of debtors, with mortgage holders more likely to be advantaged and credit card debtors facing greater financial strain.

Table 3 introduces an additional dimension by distinguishing individuals with stable versus unstable employment. Always-employed individuals not only hold more debt, average mortgage balances of $187,908 versus $105,909, but are also more likely to participate in both credit card (29.8% vs. 22.8%) and mortgage markets (52.3% vs. 34.2%). Yet this higher indebtedness does not translate into greater vulnerability: repayment stress is lower among the always-employed, with only 5.2% reporting missed mortgage payments compared to 8.9% of the not-always-employed. Health outcomes likewise show a clear gradient. The always-employed report higher health satisfaction (3.57 vs. 3.19), stronger mental health scores (MCS 74.44 vs. 69.01), and lower obesity prevalence (23.5% vs. 29.2%). These patterns suggest that employment stability is a key source of resilience, buffering households against the risks of indebtedness and enabling them to sustain better health.

Taken together, the descriptive evidence points to three key insights: (i) unsecured debt has declined sharply, while mortgage debt remains widespread and heavy; (ii) credit card and mortgage debt have distinct associations with health, the former negative and the latter neutral or slightly positive; and (iii) employment stability is an important conditioning factor. These findings lay the groundwork for the regression analysis, which formally tests whether the observed associations persist after controlling for socioeconomic and demographic factors.

### 4.2 Regression results - Debt models

Table 4 presents estimates of the relationship between household debt and health outcomes across four model specifications. Taken together, the results reveal a consistent pattern: unsecured credit card debt has the most persistent negative impact on subjective and mental health, while mortgage debt exhibits limited or inconsistent associations. Table 4 reports results in two panels: Panel A provides effect estimates in the original units of each outcome (points or percentage points), while Panel B expresses Health Satisfaction and MCS as standardized z-scores to facilitate direct comparison of effect magnitudes across outcomes.

In Panel A, the pooled OLS models indicate that the estimated effects of credit card debt are large and statistically significant. Here, coefficients represent changes in each outcome associated with a 10 percentage point increase in the corresponding debt-to-income ratio. A 10 pp increase in the credit card debt-to-income ratio is associated with a reduction of nearly 0.1 points in health satisfaction ($\beta = -0.094$, $p < 0.01$), a decline of approximately 1.1 points in the MCS ($\beta = -1.151$, $p < 0.01$), and a 4.8 percentage point higher likelihood of obesity (average marginal effect = 4.772; $\beta = 0.258$, $p < 0.01$). These magnitudes are substantial in context, given that average health satisfaction is just above 3.4 and the baseline obesity rate is 25.76%. In the OLS full-sample estimates, mortgage debt-to-income shows no detectable association with health satisfaction ($\beta = -0.001$, $p > 0.10$; or MCS ($\beta = 0.023$, $p > 0.10$, yet it is positively related to obesity, with an average marginal effect of 0.152 percentage points per 10-pp increase.

Panel B, which reports outcomes in standard deviation units, corroborates these findings: the OLS coefficient for credit card debt-to-income is −0.100 SD for health satisfaction and −0.065 SD for MCS ($p < 0.01$), while effects for mortgage debt are negligible. In fixed-effects models, standardised declines of 0.030 (health satisfaction) and 0.023 (MCS) SDs are observed. Among the always-employed, coefficients remain negative and statistically significant, underscoring that the detrimental association of unsecured debt with well-being is robust to both standardization and employment stability.

When unobserved individual characteristics are controlled for through fixed effects estimation, the adverse influence of credit card debt is reduced but remains apparent. In these models, a 10-percentage-point increase in the credit card debt-to-income ratio is associated with a significant 0.03-point reduction in health satisfaction ($\beta = -0.028$, $p < 0.05$), while the corresponding effect on the MCS is a modest 0.4-point decrease ($\beta = -0.424$, $p < 0.1$); these translate to standardised

**Table 4**. Regression results – Debt models.

| Dependent Variable | Explanatory Variable | OLS Full Sample | FE Full Sample | FE Always Employed | FE Lagged |
|---|---|---|---|---|---|
| **Panel A** | | $\beta$ (SE) | $\beta$ (SE) | $\beta$ (SE) | $\beta$ (SE) |
| Health Satisfaction | Credit debt/income | −0.094*** (0.010) | −0.028** (0.012) | −0.033** (0.015) | 0.025* (0.013) |
| | | [−0.114, −0.074] | [−0.052, −0.004] | [−0.062, −0.004] | [−0.000, 0.050] |
| | Mortgage debt/income | −0.001 (0.001) | −0.001 (0.001) | 0.000 (0.001) | −0.001 (0.001) |
| | | [−0.003, 0.001] | [−0.003, 0.001] | [−0.002, 0.002] | [−0.003, 0.001] |
| | $R^2$ | 0.134 | 0.028 | 0.023 | 0.020 |
| MCS | Credit debt/income | −1.151*** (0.200) | −0.402* (0.230) | −0.550* (0.297) | 0.059 (0.275) |
| | | [−1.543, −0.759] | [−0.853, 0.049] | [−1.132, 0.032] | [−0.480, 0.598] |
| | Mortgage debt/income | 0.023 (0.018) | 0.015 (0.018) | 0.002 (0.022) | −0.041* (0.024) |
| | | [−0.012, 0.058] | [−0.020, 0.050] | [−0.041, 0.045] | [−0.088, 0.006] |
| | $R^2$ | 0.086 | 0.011 | 0.011 | 0.013 |
| Obese | Credit debt/income | 4.772*** (0.518) | 0.0022 (0.005) | 0.0022 (0.007) | −0.002 (0.006) |
| | | [3.756, 5.788] | [−0.0076, 0.0120] | [−0.0115, 0.0159] | [−0.0138, 0.0098] |
| | Mortgage debt/income | 0.152** (0.486) | 0.0007 (0.0004) | 0.0011** (0.0005) | −0.0004 (0.0006) |
| | | [0.057, 0.248] | [−0.0001, 0.0015] | [0.0001, 0.0021] | [−0.0016, 0.0008] |
| | $R^2$ | 0.0437 (Pseudo $R^2$: Logit) | 0.035 | 0.038 | 0.026 |
| **Panel B** | | $\beta$ (SE) | $\beta$ (SE) | $\beta$ (SE) | $\beta$ (SE) |
| Health Satisfaction (z-score) | Credit debt/income | −0.100*** (0.012) | −0.030** (0.013) | −0.035** (0.016) | 0.027* (0.014) |
| | | [−0.124, −0.076] | [−0.055, −0.006] | [−0.066, −0.003] | [−0.001, 0.054] |
| | Mortgage debt/income | −0.001 (0.001) | −0.001 (0.001) | 0.000 (0.001) | −0.001 (0.001) |
| | | [−0.003, 0.001] | [−0.003, 0.001] | [−0.002, 0.003] | [−0.004, 0.002] |
| | $R^2$ | 0.1339 | 0.0282 | 0.0231 | 0.0214 |
| MCS (z-score) | Credit debt/income | −0.065*** (0.012) | −0.023* (0.013) | −0.031* (0.017) | 0.003 (0.015) |
| | | [−0.089, −0.041] | [−0.048, 0.003] | [−0.064, 0.002] | [−0.027, 0.034] |
| | Mortgage debt/income | 0.001 (0.001) | 0.001 (0.001) | 0.000 (0.001) | −0.002* (0.001) |
| | | [−0.001, 0.003] | [−0.001, 0.003] | [−0.002, 0.002] | [−0.005, 0.000] |
| | $R^2$ | 0.0856 | 0.0113 | 0.0108 | 0.0133 |

*Notes:* Coefficients ($\beta$) and standard errors (SE) are shown in parentheses, with 95% confidence intervals in brackets. Significance levels: *$p < 0.1$, **$p < 0.05$, ***$p < 0.01$. **Panel A** reports results using outcomes in their original scales. Coefficients represent changes in the outcome associated with a 10-percentage-point (pp) increase in the corresponding debt-to-income ratio. For obesity (pooled logit), entries are Average Marginal Effects (AMEs) in percentage points per 10 pp with 95% CIs. In the pooled logistic model, a 10-pp increase in the credit-card debt-to-income ratio is associated with a 4.8-pp higher probability of obesity, while the comparable effect for mortgage debt is 0.15 pp. **Panel B** reports results when continuous outcomes (Health Satisfaction and MCS) are standardised to mean 0 and SD 1. Coefficients are interpreted as standard deviation (SD) changes in the outcome per 10-pp increase in the debt-to-income ratio, allowing direct comparison of effect magnitudes across outcomes. All models include controls for age, education (years), missing-education indicator, presence of children, missing-children indicator, marital status (married, divorced, widowed), employment status (employed, unemployed, not in the labour force), private-health-insurance status, missing-insurance indicator, real household income, city residence, and survey-year fixed effects. The OLS specification additionally includes sex, country of birth, and Indigenous status, which are excluded from the fixed-effects (FE) models. Debt-to-income ratios are trimmed at 0.5 for credit-card debt and 2 for mortgage debt to reduce the influence of extreme values.
*Notes:* **FE $R^2$** magnitudes are typical for within-person panels, where limited intra-individual variation constrains the attainable $R^2$; inference emphasises coefficient sizes and robustness [17,21].

declines of 0.030 and 0.023 standard deviations, respectively. For mortgage debt, the estimated effect on health satisfaction is -0.40 points ($\beta = −0.402$, $p < 0.1$), though the association with MCS is negligible ($\beta = 0.015$). The effect of credit card debt on obesity disappears entirely once individual heterogeneity is accounted for, suggesting that cross-sectional differences in health behaviours or endowments explain much of the OLS association.

The analysis of the always-employed subsample confirms that these relationships are not confined to households experiencing unstable labour-market participation. Among continuously employed individuals, the detrimental effects of credit card debt are slightly stronger, with health satisfaction and MCS falling by 0.33 points ($\beta = −0.326$, $p < 0.05$) and 5.5 points ($\beta = −5.495$, $p < 0.1$), respectively, per 10-percentage-point increase in the credit card debt-to-income ratio.

The persistence of these effects among stably employed individuals suggests that unsecured debt burdens undermine well-being even in households with steady income streams.

Finally, when debt measures are lagged in an effort to address concerns about reverse causality, the coefficients for credit card debt become small and mostly insignificant. The one exception is a modest positive association with lagged credit card debt in the health satisfaction model ($\beta = 0.25$, $p < 0.1$), though this effect is weak and of uncertain interpretation. The absence of significant lagged effects supports the view that deteriorating health does not precede debt accumulation; rather, adverse health outcomes emerge contemporaneously with increases in unsecured debt.

Taken together, the findings from Table 4.2 indicate that unsecured debt, particularly credit card liabilities, consistently undermines subjective and mental health, while mortgage debt appears largely neutral once individual heterogeneity is controlled for. However, debt levels tell only part of the story. They capture the burden of liabilities but not the acute strain of falling behind on payments. To address this sharper form of financial distress, attention is next turned to repayment difficulties, specifically whether households report being unable to meet mortgage commitments on time.

### 4.3 Regression results - Mortgage stress

Table 5 presents the estimated effects of mortgage repayment stress, defined as households being unable to meet mortgage obligations on time. Analyses of mortgage stress are restricted to respondents who reported having a mortgage. These results highlight repayment difficulties as a distinct and acute dimension of financial strain, with consistent adverse associations for subjective and mental health outcomes, and more nuanced evidence for obesity.

In the pooled OLS specification, repayment stress emerges as a powerful predictor of diminished well-being. Households reporting difficulties record a reduction of 0.31 points in health satisfaction ($\beta = -0.312$, $p < 0.01$) and a decline of more than eight points on the MCS ($\beta = -8.336$, $p < 0.01$). These magnitudes are sizable relative to their respective scales and suggest that repayment problems impose immediate and substantial psychological and emotional costs. The OLS models indicate that mortgage stress is positively associated with obesity, but the magnitude weakens considerably in fixed effects specifications, suggesting that cross-sectional differences rather than within-individual changes may drive the OLS estimates.

**Table 5. Regression results – Could not pay mortgage.**

| Dependent Variable | Explanatory Variable | OLS Full Sample | FE Full Sample | FE Always Employed | FE Lagged |
|---|---|---|---|---|---|
| | | $\beta(SE)$ | $\beta(SE)$ | $\beta(SE)$ | $\beta(SE)$ |
| Health Satisfaction | Could not pay the mortgage | −0.312*** (0.020) | −0.081*** (0.020) | −0.096*** (0.026) | 0.019 (0.028) |
| | | [−0.351, −0.273] | [−0.121, −0.041] | [−0.147, −0.045] | [−0.036, 0.074] |
| | $R^2$ | 0.1280 | 0.0311 | 0.0257 | 0.0233 |
| MCS (SF-36) | Could not pay the mortgage | −8.336*** (0.419) | −2.884*** (0.429) | −3.412*** (0.559) | 0.495 (0.633) |
| | | [−9.158, −7.514] | [−3.725, −2.043] | [−4.508, −2.316] | [−0.747, 1.737] |
| | $R^2$ | 0.0862 | 0.0164 | 0.0152 | 0.0134 |
| Obese | Could not pay the mortgage | 5.454*** (0.887) | 0.007 (0.009) | 0.009 (0.012) | −0.028** (0.013) |
| | | [3.716, 7.192] | [−0.010, 0.024] | [−0.015, 0.033] | [−0.053, −0.003] |
| | $R^2$ | 0.0370 (Pseudo $R^2$) | 0.0404 | 0.0420 | 0.0351 |

*Notes:* Coefficients ($\beta$) and standard errors (SE) are shown in parentheses, with 95% confidence intervals in brackets. Significance levels: *$p < 0.1$, **$p < 0.05$, ***$p < 0.01$. All models control for age, education (years), a missing-education indicator, presence of children, a missing-children indicator, marital status (married, divorced, widowed), employment status (employed, unemployed, not in the labour force), private-health-insurance status, a missing-insurance indicator, real household income, city residence, and survey-year fixed effects. The OLS models (column 1) additionally include sex, country of birth, and Indigenous status, which are time-invariant and therefore excluded from the fixed-effects (FE) models. For the obesity outcome, column 1 reports Average Marginal Effects (AMEs) from a pooled logit model, expressed in percentage points (pp), to show the change in the probability of obesity for respondents who reported mortgage-repayment difficulties relative to those who did not. Columns 2–4 present results from linear probability models (LPMs) with individual and year fixed effects; these coefficients are interpreted as changes in probability.
*Notes:* **FE $R^2$** magnitudes are typical for within-person panels, where limited intra-individual variation constrains the attainable $R^2$; inference emphasises coefficient sizes and robustness [17,21].

Controlling for unobserved, time-invariant characteristics through individual fixed effects mitigates the magnitude of the estimated effects but preserves the overall pattern. Within-person changes in repayment stress are associated with a 0.08-point decline in health satisfaction ($\beta = -0.081$, $p < 0.01$) and a 2.9-point reduction in the MCS ($\beta = -2.884$, $p < 0.01$), both statistically significant and meaningful in substantive terms. These results indicate that the burden of repayment difficulties is not merely a cross-sectional artifact but corresponds to real deteriorations in well-being for the same individuals over time. In contrast, the link to obesity vanishes under the fixed effects design, suggesting that the OLS result was driven mainly by stable individual traits correlated with both repayment difficulties and health.

Restricting the analysis to the always-employed subsample yields estimates closely aligned with the full fixed effects results. Health satisfaction falls by around 0.10 points ($\beta = -0.096$, $p < 0.01$) and the MCS by 3.4 points ($\beta = -3.412$, $p < 0.01$) among those who remain continuously employed, confirming that repayment stress exerts a toll on well-being even in the absence of employment instability. This robustness across employment conditions strengthens the interpretation that repayment stress operates as an independent stressor, rather than simply reflecting broader financial or labour-market precarity.

Lagged models provide an additional test of directionality. If health deterioration were driving repayment stress, it would be expected that lagged stress predicts subsequent declines in health. Instead, the lagged coefficients are small and insignificant for most outcomes, with only a modest negative association for obesity. This pattern implies that repayment difficulties and poor health co-occur contemporaneously, but with little evidence that health shocks systematically translate into future repayment problems.

## 4.4 Robustness checks

To assess the sensitivity of the results to alternative trimming thresholds, the models were re-estimated using higher caps on the mDTI ratio – first at 6 and then at 10 (see Table 6). Focus is placed on mortgage debt because credit card debt-to-income ratios are already bounded at lower levels, whereas the distribution of mortgage balances relative to income can be extreme, making a restrictive cap of 2 potentially conservative in practice.

Across both alternative specifications, the substantive conclusions remain unchanged. Credit card debt continues to show adverse and statistically significant associations with health satisfaction and mental health, while mortgage debt remains largely neutral. Obesity results are also consistent with the baseline, with no meaningful within-person effects. While the overall pattern of results is the same, in some models the associations between credit card debt and health outcomes become even more statistically significant under the higher mDTI caps, reinforcing confidence in the robustness of these effects.

When the mortgage debt trimming threshold is relaxed to six or ten, associations for credit card debt and mental health strengthen, while associations with health satisfaction attenuate; results for the always-employed subsample remain present, which suggests that the unsecured debt relationship with mental health is the most stable feature across robustness variants. These findings indicate that a small set of extreme values does not drive the main results, and the core association between unsecured debt and mental health remains robust to alternative trimming choices.

Overall, the empirical evidence suggests that debt affects health through multiple pathways: the steady pressure of unsecured liabilities and the acute strain of repayment stress. The persistence of mental health effects, in particular, provides a foundation for the discussion that follows, in which mechanisms, policy implications, and connections to the broader literature are examined.

## 5 Discussion

This study provides new evidence on the relationship between household debt and health, highlighting the importance of distinguishing between different types of debt and repayment difficulties. The results show that unsecured credit card debt

**Table 6**. Robustness of debt–health estimates to alternative mortgage DTI specifications.

| Panel A: Mortgage DTI Trim = 6 | | | | | |
|---|---|---|---|---|---|
| **Dependent Variable** | **Explanatory Variable** | **OLS Full Sample** | **FE Full Sample** | **FE Always Employed** | **FE Lagged** |
| Health Satisfaction | Credit debt/income | −0.086*** (0.009) | −0.015 (0.009) | −0.021* (0.011) | 0.018* (0.009) |
| | | [−0.104, −0.068] | [−0.033, 0.003] | [−0.042, 0.000] | [0.000, 0.036] |
| | Mortgage debt/income | 0.001 (0.001) | −0.000 (0.000) | 0.000 (0.000) | 0.000 (0.000) |
| | | [−0.001, 0.003] | [−0.002, 0.002] | [−0.002, 0.002] | [−0.002, 0.002] |
| | $R^2$ | 0.125 | 0.027 | 0.022 | 0.021 |
| MCS (SF-36) | Credit debt/income | −1.295*** (0.168) | −0.396** (0.182) | −0.501** (0.226) | 0.121 (0.201) |
| | | [−1.624, −0.966] | [−0.753, −0.039] | [−0.944, −0.058] | [−0.273, 0.515] |
| | Mortgage debt/income | 0.014** (0.006) | 0.003 (0.006) | 0.005 (0.007) | −0.006 (0.007) |
| | | [0.002, 0.026] | [−0.008, 0.014] | [−0.009, 0.019] | [−0.020, 0.008] |
| | $R^2$ | 0.078 | 0.012 | 0.011 | 0.014 |
| Obese | Credit debt/income | 4.529*** (0.442) | −0.001 (0.004) | 0.002 (0.005) | −0.001 (0.005) |
| | | [3.662, 5.396] | [−0.009, 0.007] | [−0.008, 0.012] | [−0.011, 0.009] |
| | Mortgage debt/income | −0.017 (0.017) | 0.000 (0.000) | 0.000 (0.001) | −0.000 (0.001) |
| | | [−0.050, 0.017] | [−0.001, 0.001] | [−0.001, 0.002] | [−0.002, 0.001] |
| | $R^2$ | 0.0415 (Pseudo $R^2$) | 0.037 | 0.039 | 0.027 |
| **Panel B: Mortgage DTI Trim = 10** | | | | | |
| Health Satisfaction | Credit debt/income | −0.087*** (0.009) | −0.015* (0.009) | −0.022** (0.011) | 0.017* (0.010) |
| | | [−0.105, −0.069] | [−0.033, 0.003] | [−0.043, −0.001] | [−0.003, 0.037] |
| | Mortgage debt/income | 0.000 (0.000) | −0.000 (0.000) | 0.000 (0.000) | 0.000 (0.000) |
| | | [−0.001, 0.003] | [−0.002, 0.002] | [−0.002, 0.002] | [−0.002, 0.002] |
| | $R^2$ | 0.125 | 0.027 | 0.022 | 0.021 |
| MCS (SF-36) | Credit debt/income | −1.320*** (0.165) | −0.458** (0.179) | −0.571** (0.224) | 0.090 (0.194) |
| | | [−1.643, −0.997] | [−0.809, −0.107] | [−1.010, −0.132] | [−0.291, 0.471] |
| | Mortgage debt/income | 0.012** (0.005) | −0.003 (0.005) | −0.001 (0.006) | −0.001 (0.006) |
| | | [0.002,0.022] | [−0.013, 0.007] | [−0.013,0.011] | [−0.013,0.011] |
| | $R^2$ | 0.077 | 0.012 | 0.011 | 0.014 |
| Obese | Credit debt/income | 4.422*** (0.432) | 0.000 (0.004) | 0.003 (0.005) | −0.003 (0.005) |
| | | [3.576, 5.268] | [−0.008, 0.008] | [−0.007, 0.013] | [−0.013, 0.007] |
| | Mortgage debt/income | −0.027* (0.014) | −0.000 (0.000) | 0.000 (0.001) | −0.000 (0.001) |
| | | [−0.055, 0.001] | [−0.001, 0.001] | [−0.001, 0.002] | [−0.002, 0.002] |
| | $R^2$ | 0.0415 (Pseudo $R^2$) | 0.037 | 0.039 | 0.027 |

*Notes:* Coefficients ($\beta$) and standard errors (SE) are shown in parentheses, with 95% confidence intervals in brackets. Significance levels: *$p < 0.1$, **$p < 0.05$, ***$p < 0.01$. All coefficients are rescaled such that one unit equals a 10-percentage-point increase in the debt-to-income ratio. Results include the same control variables as Table 4. Panels A and B present robustness checks for alternative mortgage-DTI trimming thresholds (6 and 10). The coefficients remain qualitatively similar to the baseline, indicating that outliers do not materially influence the estimates. For obesity, the pooled logit model reports average marginal effects (AMEs) in percentage points (per 10 pp).
*Notes:* FE $R^2$ magnitudes are typical for within-person panels, where limited intra-individual variation constrains the attainable $R^2$; inference emphasises coefficient sizes and robustness [17,21].

is consistently associated with poorer subjective and mental health outcomes, whereas mortgage debt, despite being the most significant household liability, exhibits minimal direct effects on these outcomes. These findings reinforce the idea that the structure and purpose of debt matter as much as its magnitude. Credit card liabilities typically carry high interest rates and are often accumulated during periods of financial distress, creating persistent psychological burdens. By contrast, mortgage debt is asset-backed, carries lower interest rates, and is linked to the benefits of home ownership, such as stability and wealth accumulation, which may offset its potential stress effects.

The regression estimates demonstrate that a one-unit increase in the credit card debt-to-income ratio is associated with within-person declines of around 0.3 points in health satisfaction and 4-5 points in mental health scores. These magnitudes are large relative to baseline levels and indicate a significant deterioration in well-being when unsecured debt burdens increase. Importantly, these effects remain in fixed-effects models, which control for all stable individual characteristics, suggesting that differences across households do not merely drive the results. This pattern is consistent with earlier

research linking unsecured debt to diminished well-being [17,19,21,41] and extends that literature by showing that even stock measures of credit card balances, in the absence of reported repayment problems, exert meaningful adverse health effects.

Presenting effects in standard deviation units confirms that unsecured credit burdens are most consequential for mental health; estimated changes in the MCS per 10 percentage point increase in the credit card ratio translate to roughly two to three percent of a standard deviation within persons, with consistent patterns in the always employed subsample. Robustness appears stronger for mental health than for health satisfaction once extreme mortgage balances are accommodated through higher trimming thresholds.

The analysis of repayment stress further highlights how acute financial difficulties can impact health. Inability to meet mortgage payments is associated with declines of around 0.08-0.10 points in health satisfaction and reductions of roughly three points in the MCS in fixed-effects models, magnitudes that are both statistically significant and clinically meaningful. These findings echo studies documenting the severe consequences of foreclosure and housing insecurity [33,34], but extend them by showing that even self-reported repayment difficulties short of foreclosure carry substantial health costs. While OLS models suggested a higher likelihood of obesity among those experiencing repayment stress, these associations vanish in fixed-effects estimation, reinforcing that the strongest and most credible effects are concentrated in the psychological domain.

The temporal dynamics explored through lagged models add further nuance. For credit card debt, contemporaneous effects are stronger than lagged ones, suggesting that the psychological burden of unsecured debt manifests quickly rather than gradually. For mortgage repayment stress, lagged coefficients are small and generally statistically insignificant, indicating that repayment difficulties exert their influence acutely, with immediate effects on mental well-being rather than delayed consequences. Together, these results point to distinct mechanisms: unsecured debt imposes a chronic toll that accumulates over time, while repayment crises trigger sharp, short-term declines in health.

The absence of robust within-person effects on obesity, despite consistent associations for subjective health and mental health, indicates that debt primarily affects psychological rather than physical health in the short to medium term. This finding resonates with evidence that stress, anxiety, and a loss of control are central pathways linking financial strain and well-being [22,30]. While longer-term physical consequences cannot be ruled out, the regression evidence suggests that the immediate burden of debt is felt most strongly in the mental health domain.

The regression results also highlight the role of employment stability. Estimates restricted to the always-employed subsample (Table 4) indicate that the adverse effects of credit card debt persist even among households with continuous labour market attachment. In these models, a 10-percentage-point increase in credit card debt-to-income is associated with a 0.033-point reduction in health satisfaction ($\beta = -0.0326$, $p < 0.05$) and a 0.55-point decline in mental health composite score ($\beta = -0.5495$, $p < 0.1$). Standardized estimates from Panel B confirm these effects, corresponding to declines of 0.035 and 0.031 standard deviations in health satisfaction and MCS, respectively. By contrast, mortgage debt-to-income continues to show no statistically significant association with either outcome in this subsample (all coefficients near zero, $p>0.10$). These results suggest that employment stability confers some resilience against financial shocks, but does not fully insulate households from the well-being costs of unsecured debt burdens.

From a policy perspective, addressing the health implications of household debt requires measures grounded in the underlying mechanisms through which debt exerts its effects. Chronic financial strain activates stress-related biological and psychosocial pathways, while material constraints limit access to health-promoting resources such as nutrition and preventive care. Accordingly, preventive strategies should prioritise the regulation of high-cost consumer credit through affordability assessments and interest rate controls, coupled with expanded access to affordable short-term lending. Integrating financial counselling and debt management support into primary and mental healthcare services may attenuate the cumulative psychological burden associated with unsecured debt. Crisis interventions that provide temporary mortgage relief or restructuring assistance are particularly relevant given the evidence of distress linked to repayment difficulties. Policies promoting employment stability and income security indirectly enhance health outcomes by reducing

exposure to debt-related stressors. Although these recommendations are supported by longitudinal associations, residual confounding by time-varying factors such as job loss or acute health events cannot be entirely excluded; the findings therefore indicate robust associations rather than definitive causal effects.

Several limitations should be acknowledged. The analysis relies on self-reported health measures, including subjective health satisfaction and BMI, which may be subject to reporting errors. While fixed-effects models mitigate concerns about unobserved heterogeneity, reverse causality cannot be entirely excluded, particularly for mental health outcomes. Debt measures are based on self-reported balances and do not capture repayment schedules, interest rates, or liquidity constraints, which could further shape financial stress. Finally, the sample is limited to working-age Australians, and future research could extend to older populations, younger adults, or cross-country comparisons.

Despite these caveats, this study provides the first longitudinal evidence on debt and health in Australia, a country with some of the highest household debt-to-income ratios in the OECD. By disaggregating debt types and incorporating measures of repayment stress, the results show that unsecured debt and acute financial crises carry significant health burdens, while mortgage debt in itself appears less harmful. These insights highlight the need for targeted policies that address both chronic and acute forms of financial strain, recognising household debt as a critical factor shaping population health.

## 6 Conclusion

This study, to the best of our knowledge, provides the first comprehensive Australian evidence on the health effects of household debt, showing that unsecured credit card debt consistently undermines health satisfaction and mental health, while mortgage debt has negligible impacts despite its larger financial burden. By distinguishing between chronic debt exposure and acute repayment stress, the findings reveal that the type and structure of debt, rather than its size alone, are critical determinants of well-being. The strong associations between mortgage repayment difficulties and deteriorating mental health underscore the severe toll of acute financial distress, while the robustness of results across fixed-effects, always-employed, and lagged models increases confidence in their validity. Taken together, the evidence highlights the need for targeted policy interventions such as tighter consumer credit protections, early support for households in repayment difficulty, and integrated approaches linking financial and health assistance, while also pointing to the importance of further research into longer-term health consequences and subgroup vulnerabilities. For policymakers, the message is clear: addressing unsecured credit and repayment stress is not only an economic imperative but also a public health priority.

## Author contributions

**Conceptualization:** Khaled Toffaha, Michael A. Kortt, Albert Wijeweera.

**Data curation:** Michael A. Kortt.

**Formal analysis:** Khaled Toffaha, Michael A. Kortt, Albert Wijeweera.

**Investigation:** Michael A. Kortt, Albert Wijeweera.

**Methodology:** Maram Tammam, Khaled Toffaha, Michael A. Kortt, Albert Wijeweera.

**Supervision:** Michael A. Kortt, Albert Wijeweera.

**Visualization:** Maram Tammam, Khaled Toffaha.

**Writing – original draft:** Maram Tammam, Khaled Toffaha, Michael A. Kortt.

**Writing – review & editing:** Albert Wijeweera.

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
