## [Decision Letter · Decision Letter 0]

9 Oct 2025

PONE-D-25-48109Household Indebtedness and Well-being: Evidencefrom AustraliaPLOS ONE

Dear Dr. Toffaha,

Thank you for submitting your manuscript to PLOS ONE. After careful consideration, we feel that it has merit but does not fully meet PLOS ONE’s publication criteria as it currently stands. Therefore, we invite you to submit a revised version of the manuscript that addresses the points raised during the review process.

After reviewing your study as well as the reviews of two qualified experts, I recommend that you revise your manuscript according to the suggestions of the reviewers. Both reviewers offered relatively minor feedback on suggestions that you should make, and I agree with their suggestions to improve the readability of your manuscript and results. Additionally, in your regression results, I suggest adding metrics to express the overall model fit to give the reader a better since of how important your key variables are to explaining variation in outcomes. After making the revisions suggested by the referees I would be happy to reconsider your manuscript for publication.

We look forward to receiving your revised manuscript.

Kind regards,

Meagan McCollum

Academic Editor

PLOS ONE

Journal Requirements:

3. Thank you for uploading your study's underlying data set. Unfortunately, the repository you have noted in your Data Availability statement does not qualify as an acceptable data repository according to PLOS's standards.

Reviewer's Responses to Questions

**Comments to the Author**

1. Is the manuscript technically sound, and do the data support the conclusions?

Reviewer #1: Yes

Reviewer #2: Yes

2. Has the statistical analysis been performed appropriately and rigorously?

Reviewer #1: I Don't Know

Reviewer #2: I Don't Know

3. Have the authors made all data underlying the findings in their manuscript fully available?

Reviewer #1: No

Reviewer #2: No

4. Is the manuscript presented in an intelligible fashion and written in standard English?

Reviewer #1: Yes

Reviewer #2: Yes

5. Review Comments to the Author

Reviewer #1: Dear Authors,

Thank you for the opportunity to read your work. The manuscript is clearly written, easy to follow, and the main contributions are well articulated. i noted a few minor issues that could improve clarity.

General notes:

You noted that cDTI is trimmed at 0.5, yet in the abstract/text the FE coefficient moves from 0  1 ( i.e 0% to 100%), which lies outside the observed range. It would be clearer to report the effects per 10 percentage point (pp). Currently the coefficient is reported per 1.0 (i.e., per 100 pp). However, if you change the effect per 10 pp, you estimate the coefficient × 0.10. The following results would occur for Table 4: Health satisfaction FE : −0.284 per 1.0  −0.028 per 10 pp.; MCS FE: −4.024 per 1.0  −0.402 per 10 pp.; Obesity FE (LPM): 0.022 per 1.0 +0.0022 per 10 pp (that’s +0.22 pp). Either I would adjust the trimmed threshold or would restate the effects statements accordingly.

It would be helpful to report the effect sizes in SD units, so readers can compare magnitudes across outcomeson a common scale.

Specific notes:

Part 4.2 Regression Results

In Table 4, the pooled obesity model is logistic; thus, the 2.585 coefficient is on the log-odds scale, not percentage points. Please report average marginal effects (AMEs) with 95% CIs (ideally per 10pp change) to express results in percentage points.

Part 4.3 Mortgage Stress

Table 5 appears to treat non-mortgage households as having no mortgage stress.

Problem: "Couldn’t pay the mortgage" is only meaningful for people who have a mortgage. If you include people without a mortgage and code their stress as 0, you’re treating “no mortgage” the same as "has a mortgage and no stress"

To avoid exposure misclassification, please either restrict the sample to mortgage holders or include a mortgage-holder indicator and its interaction with the stress variable, and state this explicitly in the table notes.

Part 4.4 Robustness Check

When relaxing the mDTI trim to 6 or 10 (Table 6), the FE cDTI association with MCS strengthens, while the association with health satisfaction weakens. A more nuanced conclusion seems appropriate: Robustness is strong for mental health and weaker for health satisfaction, with effects remaining in the always-employed subsample.

Kind regards

Reviewer #2: I responded "I Don't Know" to Question #2 (Has the statistical analysis been performed appropriately and rigorously?) because I am not qualified to assess the authors’ statistical methods. For this review, I assumed that the analyses were appropriately designed and executed.

Regarding my “No” response to Question #3 (Have the authors made all data underlying the findings in their manuscript fully available?), the authors explained why this was not possible in the cover document.

This manuscript describes a thoughtfully conceived study that deepens understanding of financial debt as a risk factor for poor health. The authors effectively explained the rationale for their investigation within the context of prior research regarding associations between indebtedness and health. This includes their rationale for focusing on credit card debt, mortgage debt, and occasional inability to make mortgage payments. Their arguments regarding how this study adds to prior research are clear, both in terms of the specific questions they address and why a focus on Australia is distinctive. The study data are derived from five successive waves of a national health survey conducted over a 14-year period and represent substantial numbers of respondents. Their outcome measures (overall health status, mental health status, and obesity) are derived from previously-validated survey questions. Their results are clearly explained and presented. I especially appreciate the presentation and discussion of descriptive findings (Tables 1-3 and accompanying text) in advance of the presentation of modelling results, which grounds and contextualizes the modeling results. In presenting their model findings, they “walk the reader through” the results in a way that successfully explains and interprets the modelling results, both in the results section itself and in summarizing their findings in the discussion section.

I have a few minor recommendations.

It would be helpful to define the term “stock measures” of household liability when the term is first used in the background section, even though the reader can infer its meaning from the subsequent description of debt measures in the methods section.

There appears to be missing text at the start of the 2nd full paragraph on page 2, i.e., the name of the author(s) referred to in the statement, “Building on [17], who used German panel data…”

Tables 4-6 could be more clearly labelled. In their present formats, the reader is entirely dependent on the accompanying text to understand them, and it is not clear what the values in parentheses represent.

6. PLOS authors have the option to publish the peer review history of their article (what does this mean?). If published, this will include your full peer review and any attached files.

Reviewer #1: No

Reviewer #2: No

---

## [Author Response · Author response to Decision Letter 1]

24 Oct 2025

Response to Reviewer 1

Comment 1.1

Rescale the effects so that results are stated per ten percentage point change...

Response:

The debt to income ratios for credit card and mortgage balances were divided by ten prior to estimation; all coefficients and average marginal effects now correspond to a ten percentage point increase. Methods state this respecification; Results and the Abstract have been edited to ensure consistent interpretation. The revised Table 4 also includes a standardised panel in which health satisfaction and the Mental Component Score are transformed to z scores; the text reports these standard deviation units where appropriate.

Comment 1.2

For the pooled obesity model, the coefficients reported previously were on the log odds scale. Report average marginal effects with ninety five percent confidence intervals, preferably per ten percentage point change.

Response:

The pooled obesity specification now reports average marginal effects in percentage points per ten percentage point increase in the debt to income ratio; 95% confidence intervals appear in the tables. The table notes and the Methods section specify the estimand and the presentation.

Comment 1.3

Avoid exposure misclassification in the mortgage stress models...

Response:

All mortgage stress models are restricted to respondents with an active mortgage as mentioned (“Could not pay the mortgage”); the Methods section states this analytic population and Table 5 notes the sample restriction explicitly. The Results subsection on mortgage stress opens with a sentence that reiterates this restriction.

Comment 1.4

Temper the robustness claims when relaxing the mortgage trimming threshold...

Response:

The robustness subsection has been revised to adopt the suggested nuance. The text now states that fixed-effects associations are most stable for the Mental Component Score, with weaker and less stable patterns for health satisfaction once mortgage trimming is relaxed; the always employed subsample yields consistent patterns for mental health.

Comment 1.5

Clarify the confidence interval and unit conventions in the tables.

Response:

All regression tables include harmonised notes that state the unit of the coefficients, that parentheses contain standard errors, and that brackets contain ninety five percent confidence intervals. These notes also document the rescaling by ten percentage points and the use of average marginal effects for the pooled obesity models.

Response to Reviewer 2

Comment 2.1

Define the term “stock measures” at first use in the background section.

Response:

The Introduction now defines stock measures at first use as outstanding balances recorded at the interview date that capture the level of liabilities rather than flows such as missed repayments; this clarifies the conceptual distinction that motivates the empirical strategy.

Comment 2.2

Repair the sentence that begins “Building on [17]” by naming the author or authors.

Response:

The sentence now reads: “Building on Keese and Schmitz, who used German panel data to demonstrate that repayment difficulties across consumer credit and mortgage liabilities undermine health, this study examines stock measures of debt in an Australian context.”

Comment 2.3

Strengthen table titles and notes so that a reader can understand units and notation without referring back to the text.

Response:

Tables now identify the unit of change for the regressors, the model family, and whether outcomes are standardised; table notes specify that parentheses contain standard errors and brackets contain ninety five percent confidence intervals, that coefficients for debt to income ratios represent changes per ten percentage point increase, and that pooled logit results for obesity are presented as average marginal effects in percentage points; fit measures are reported as described above.

---

## [Editor Report · Decision Letter 1]

4 Nov 2025

Household Indebtedness and Well-being: Evidencefrom Australia

PONE-D-25-48109R1

Dear Dr. Toffaha,

We’re pleased to inform you that your manuscript has been judged scientifically suitable for publication and will be formally accepted for publication once it meets all outstanding technical requirements.

Kind regards,

Meagan McCollum

Academic Editor

PLOS ONE
---

## [Editor Report · Acceptance letter]

PONE-D-25-48109R1

PLOS ONE

Dear Dr. Toffaha,

I'm pleased to inform you that your manuscript has been deemed suitable for publication in PLOS ONE. Congratulations! Your manuscript is now being handed over to our production team.

Kind regards,

on behalf of

Dr. Meagan McCollum

Academic Editor

PLOS ONE